# Malignancy Prediction Capacity and Possible Prediction Model of Circulating Tumor Cells for Suspicious Pulmonary Lesions

**DOI:** 10.3390/jpm11060444

**Published:** 2021-05-21

**Authors:** Ching-Yang Wu, Jui-Ying Fu, Ching-Feng Wu, Ming-Ju Hsieh, Yun-Hen Liu, Hui-Ping Liu, Jason Chia-Hsun Hsieh, Yang-Teng Peng

**Affiliations:** 1Thoracic and Cardiovascular Surgery Division, Department of Surgery, Chang Gung Memorial Hospital, Linkou 333423, Taiwan; wu.chingyang@gmail.com (C.-Y.W.); maple.bt88@gmail.com (C.-F.W.); hsiehmj2@cgmh.org.tw (M.-J.H.); l571011l@cgmh.org.tw (Y.-H.L.); tpeclcra@gmail.com (H.-P.L.); 2Department of Medicine, Medical College, Chang Gung University, Linkou 333323, Taiwan; juiing0917@gmail.com; 3Pulmonary and Critical Care Medicine, Department of Internal Medicine, Chang Gung Memorial Hospital, Linkou 333423, Taiwan; 4Division of Hematology-Oncology, Department of Internal Medicine, New Taipei Municipal Tu-Cheng Hospital, New Taipei City 236017, Taiwan; 5Division of Hematology-Oncology, Department of Internal Medicine, Chang Gung Memorial Hospital, Linkou 333423, Taiwan; 6Institute of Epidemiology and Preventive Medicine, College of Public Health, National Taiwan University, Taipei 100025, Taiwan; k74866740@gmail.com

**Keywords:** circulating tumor cell, malignancy prediction, pulmonary lesion

## Abstract

More and more undetermined lung lesions are being identified in routine lung cancer screening. The aim of this study was to try to establish a malignancy prediction model according to the tumor presentations. From January 2017 to December 2018, 50 consecutive patients who were identified with suspicious lung lesions were enrolled into this study. Medical records were reviewed and tumor macroscopic and microscopic presentations were collected for analysis. Circulating tumor cells (CTC) were found to differ between benign and malignant lesions (*p* = 0.03) and also constituted the highest area under the receiver operation curve other than tumor presentations (*p* = 0.001). Since tumor size showed the highest sensitivity and CTC revealed the best specificity, a malignancy prediction model was proposed. Akaike information criterion (A.I.C.) of the combined malignancy prediction model was 26.73, which was lower than for tumor size or CTCs alone. Logistic regression revealed that the combined malignancy prediction model showed marginal statistical trends (*p* = 0.0518). In addition, the 95% confidence interval of combined malignancy prediction model showed less wide range than tumor size ≥ 0.7 cm alone. The calculated probability of malignancy in patients with tumor size ≥ 0.7 cm and CTC > 3 was 97.9%. By contrast, the probability of malignancy in patients whose tumor size was < 0.7 cm, and CTC ≤ 3 was 22.5%. A combined malignancy prediction model involving tumor size followed by the CTC count may provide additional information to assist decision making. For patients who present with tumor size ≥ 0.7 cm and CTC counts > 3, aggressive management should be considered, since the calculated probability of malignancy was 97.9%.

## 1. Introduction

Lung cancer is a leading cause of cancer death worldwide. With regard to lung cancer survival, patients detected in the early stages have a better prognosis. However, they are difficult to identify, since there are no obvious clinical symptoms and signs. Therefore, lung cancer screening was implemented in order to identify early stage lung cancer patients. The National Lung Screening Trial determined that low-dose computed tomography (LDCT) decreased the relative risk of lung-cancer-related death by 20%, compared with chest plain film [1]. However, LDCT only detects the difference in radio density between lesions, which could be benign or malignant, and surrounding lung parenchyma. As a result, in the era of lung cancer screening, as more LDCT’s have been done, more undetermined pulmonary lesions have been identified. Hence, it has become crucial for clinical practitioners to be able to pick out those lesions, which really need to be managed. Many studies [2,3] and guidelines [4,5,6] recommend using tumor size as presented in chest tomography as a basis for deciding the optimal timing of tissue proving. However, no specific tumor size has been recommended. Other image characteristics, such as consolidation–tumor ratio (C/T ratio) [7] and standard uptake value (SUV) [8,9,10] of the tumor, have also been investigated to differentiate the nature of suspected lesions. From the literature review, C/T ratio can vary with different slice thicknesses [11,12] and tumor SUV might not be detected in tumors of size less than 1 cm or in certain tumor cell types, such as neuroendocrine tumors or lepidic adenocarcinoma with low glucose metabolic capacity [13]. No specific image characteristics can be used as malignancy predictors for undetermined lung lesions.

Many tumor markers have also been investigated for malignancy prediction capacity [14,15]. However, no biomarker has qualified as a single diagnostic criterion for malignancy prediction due to insufficient sensitivity and specificity with low reproducibility [15,16]. This may be related to the heterogeneity of tumor cells, which have varying exocrine capacity. In order to overcome this and improve the prediction capacity, multi-biomarker combination prediction models have been proposed and investigated [17,18]. However, no definite combination model for malignancy prediction has been identified and recommended. Based on the pathogenesis of cancer metastases, tumor cells break down the basement membrane and progress from benign carcinoma in situ to malignant invasive cancer. Once tumor cells invade through the basement membrane, they may migrate toward vessels and penetrate the vascular wall into systemic circulation [19,20]. As a result, tumor cells and their breakdown materials can be detected in blood samples [21]. Not only circulating tumor cells but also breakdown materials, such as circulating tumor deoxynucleic acid (ct DNA) and micro ribonucleic acid (MicroRNA) have been correlated to prognosis [22,23,24] and treatment response [25,26,27]. Some limited evidence has shown that CTCs can potentially be employed in a combination role for lung cancer diagnosis [28]. In addition, folate receptor-positive CTCs have been proven as a useful diagnostic tool for lung cancer [29,30]. Taken together, these results expose the possible correlations between CTCs and lung cancer, and the potential capacity for malignancy prediction. From the view of primary cancer, tumor cells may detach and enter the blood stream once they have invaded through the basement membrane. From the view of extra-pulmonary malignancy of lung metastases, tumor cells can be identified, because they spread from the primary site to the lung parenchyma. In this study, we attempt to determine the malignancy prediction capacity of CTCs and propose a possible combined malignancy prediction model composed of macroscopic and microscopic presentations.

## 2. Materials and Methods

### 2.1. Patients and Enrollment Criteria

From January 2017 to December 2018, 50 consecutive patients with suspicious lung lesions, including lung nodule greater than 1 cm and increased C/T ratio among tumor, which measured between 0.5 to 1 cm, were recruited into this study. Only patients whose ages were between 20–90 years agreed to receive tumor resection for tissue proof and regular post-operation surveillance were enrolled. (Appendix A) All enrolled patients received complete pre-operation evaluation and tumor resection for tissue proving. CTCs were also obtained before the operation. Benefit and risk were explained before enrollment, and written informed consent was obtained. This study was approved by the Institutional Review Board under the I.R.B. numbers 201701892B0C102, 201801475B0C102, and 2019011996B0.

### 2.2. Tumor Presentation

#### 2.2.1. Macroscopic Presentation

Macroscopic presentations, i.e., tumor image characteristics, were obtained by chest computed tomography (CT) and positron emission tomography–computed tomography (PET-CT). Three image characteristics were collected, including tumor size, C/T ratio, and tumor SUV. Tumor size and C/T ratio were measured in images of 3.75 mm slide thickness using a picture archiving and communication system (PACS). These measurements represent the lesion size and the radio densities, as they correlate to pathologic findings [31]. SUV of tumors from PET-CT was measured to identify tumor glucose metabolic capacity, which represents the growth advantage observed in human cancer [32].

#### 2.2.2. Microscopic Presentation

Two categories of microscopic presentations, namely proteins secreted by the tumor and by the CTCs, were checked. Proteins secreted by the tumor, including carcinoembryonic antigen (CEA) and squamous cell carcinoma antigen (SCC), and by CTCs were checked pre-operatively via peripheral blood sampling.

#### 2.2.3. Cutoff Value of Tumor Presentation

The cutoff value of the macroscopic and microscopic tumor presentations was based on literature review. Macroscopic tumor presentation consisted of measurements, including tumor size, C/T ratio, and SUV. Cutoff values of tumor size, C/T ratio, and SUV were 0.7 cm [2], 50% [7], and 2.5 [8,9,10], respectively. Microscopic tumor presentation consisted of CEA, SCC, and CTC values. Cutoff values of CEA, SCC, and CTCs were 3.4 ng/mL [14], 3.5 ng/mL [15], and 3 [25,33,34], respectively.

### 2.3. Pre-Operative Evaluation, Operation, and Surveillance

Resectability evaluation was done by CT, PET-CT, and brain magnetic resonance image (MRI). Pulmonary function test (PFT) and cardiac echo were done for cardiopulmonary reserve evaluation. Sublobar resection, such as wedge resection and segmentectomy, were done first for tissue proving. If malignancy was confirmed, patients with good cardiopulmonary reserve received curative anatomic resection and mediastinal lymph node dissection. For those with compromised cardiopulmonary reserve, only additional mediastinal lymph nodes were done after sublobar resection. If benign lesion was confirmed, patients received sublobar resection without further mediastinal lymph node dissection, whereby surveillance and management were based on final pathology. For patients with confirmed benign lesion, such as hamartoma, regular surveillance was recommended. For patients with identified infections, anti-infection treatment was given according to culture and pathologic result. For patients diagnosed with primary lung cancer, further treatment was arranged, according to the tumor stage. For patients with noted extra-pulmonary malignancy and lung metastasis, further palliative treatment was given for systemic disease control.

### 2.4. Measurement of Circulating Tumor Cells (CTCs)

Combined negative and positive selection, which had been designed and validated [35], was utilized for CTC isolation. Four milliliters (mL) of blood were used for CTC counting, with a further 4 mL used for quality control. The negative selection, i.e., enrichment, included depletion of red blood cells (RBCs) and white blood cells (WBCs). RBC depletion was done by lysis of RBCs within 24 h after sampling. WBC depletion was done by adding EasySep CD45 Depletion Cocktail (STEMCELL Technologies Inc., Vancouver, BC, Canada) at 25 μL/mL and EasySep Magnetic Nanoparticles (STEMCELL Technologies Inc., Vancouver, BC, Canada) at 50 μL/mL and Hoechst 33,342 (1:500 in washing solution; Thermo Scientific, Waltham, MA, USA) for the nuclear staining. The positive selection, i.e., purification, was done by EpCAM isotyping. CTCs were defined as cells that were negative for CD45 and positive both for epithelial cell adhesion molecule (EpCAM) and Hoechst. Flow cytometry using CytoFLEX flow cytometer (Beckman Coulter, San Diego, CA, USA) quantitatively identified CTCs and calculated their numbers.

### 2.5. Statistics

Descriptive statistics for continuous variables are expressed as mean ± standard deviation (SD) and categorical variables are expressed as numbers (percentages). The Kruskal–Wallis test and Mann–Whitney U test were utilized to analyze the correlation between patients with malignant lesions and those with benign lesions. Receiver operating characteristic (ROC) curve was utilized to analyze the correlation between tumor presentation and tumor nature. Positive predictive and negative predictive rates among tumor presentations were calculated based on cutoff values derived from the literatures. Akaike information criterion (AIC) and logistic regression were also utilized for malignancy prediction model selection. All reported *p*-values are two sided and considered significant at *p* < 0.05. In addition, 95% confidence intervals are reported. All statistical analyses were performed using S.A.S. version 9.0.

## 3. Results

### 3.1. Characteristics of Cohort

Fifty consecutive patients were enrolled in this study, and the characteristics are shown in Table 1. Male patients constituted 54% of the cohort, and the age of the study cohort was 64.0 ± 12.4 years. The majority of the patients (48/50, 96%) had good general performance status, and 30% had past malignancy history. The macroscopic tumor presentations included tumor size, C/T ratio as measured in CT, and SUV as measured in PET-CT. The tumor sizes and C/T ratios were 2.3 ± 1.2 cm and 0.7 ± 0.4%, respectively. Tumor composition was classified into four groups, including pure ground glass opacity (GGO), GGO predominant, solid predominant, and pure solid. Twenty-four percent of the patients were identified as pure GGO and GGO predominant, while 72% patients were identified as solid predominant and pure solid lesion. C/T ratio could not be calculated in two patients who presented with cavitary lesion on CT. Tumor SUV measured by PET-CT was 6.7 ± 5.3. Microscopic tumor presentations included CEA, SCC, and CTCs. Mean CEA, SCC, and CTCs were 3.1 ± 3.3 ng/mL, 1.1 ± 0.6 ng/mL, and 12.1 ± 14.8, respectively. Ninety-two percent of patients received anatomic resection, including segmentectomy and lobectomy, with mediastinal lymph node dissection. Pathology of all resected specimens showed 92% of patients had malignant etiology, and tumor size was 2.3 ± 1.3 cm.

### 3.2. Presentations between Benign and Malignant Pulmonary Lesions

Three clinical scenarios are encountered in clinical practice, including benign tumor, pulmonary metastasis from extra pulmonary malignancy, and primary lung cancer. Among all macroscopic and microscopic tumor presentations (Table 2), patients with benign lesions had lower CTC (2.0 ± 2.5), compared with those who had extra-pulmonary malignancy with lung metastasis (17.4 ± 16.2) and primary lung cancer (12.5 ± 14.7). The intergroup difference did not reach statistical significance (*p* = 0.08). From the view of the etiology of lesions, i.e., benign versus malignant, the difference in CTC was statistically significant (2.0 ± 2.8 versus 13.0 ± 15.1, *p* = 0.03, Table 2). The predictive capacity of tumor presentation was analyzed by a receiver operating characteristic curve. CTCs revealed the highest area under the receiver operating characteristic curve other than tumor presentations (Figure 1, area under curve (AUC): 0.8261, cutoff value: 6.7, *p* = 0.001). In addition, tumor size showed the highest sensitivity, and CTC revealed the best specificity.

### 3.3. Malignancy Prediction Capacity of Tumor Presentations and Proposed Prediction Model

An ideal malignancy prediction model should have good sensitivity and specificity. We further analyzed sensitivity, specificity, positive prediction rate, and negative prediction rate by calculating them based on cutoff value, following a literature review (Table 3) [2,7,8,9,10,14,15,33,34]. Tumor size was found to have the highest sensitivity (0.96), while CTC showed best specificity (0.75); this finding was similar to our cohort. Based on these findings, we proposed that CTCs could be combined with tumor size to develop a combined malignancy prediction model, i.e., a stratification screen based on tumor size and CTCs. Hence, logistic regression was performed for this combined malignancy prediction model (Table 4, Appendix A). Akaike information criterion (A.I.C.) of the combined malignancy prediction model was 26.73, which was lower than tumor size and CTCs alone. This result may imply better goodness of fit of the combined malignancy prediction model (Table 4a). Base on the finding of A.I.C, we utilized logistic regression for model selection. Logistic regression analysis showed CTCs ≥ 3 carried higher risk of malignancy. Although the large CIs indicated an unstable model due to the benign group’s small sample, the statistical significance remains positive. (*p* = 0.038) However, a tumor size ≥ 0.7 cm did not reveal any correlation with malignancy (*p* = 0.077). The combined malignancy prediction model showed marginal statistical trends (*p* = 0.051, Table 4b). The combined malignancy prediction model showed marginal statistical trends (*p* = 0.051, Table 4b). In addition, the 95% confidence interval of the combined malignancy prediction model showed a less wide range than tumor size ≥ 0.7 cm alone (0.98 to 143.22. versus 0.74 to 303.74). From the view of probability of malignancy, the calculated probability was 97.9% in patients whose tumor size and CTCs were greater than the cutoff value. By contrast, the probability of malignancy was 22.5% in patients whose tumor size and CTCs were less than the cutoff value. Calculated malignancy probabilities for patients with greater tumor size or CTC were 79.7 and 77.5%, respectively.

## 4. Discussion

More and more undetermined lung lesions are being identified in routine lung cancer screening. Many recommendations within guidelines have been based on tumor image findings, such as tumor size, C/T ratio, and glucose metabolic capacity [2,3,4,5,6,7,8,9,10]. However, there has been no single definite recommendation because of the limitations of images and variations in patients [11,12,13]. To address this problem, tumor markers, such as CEA and SCC have undergone further investigation, but no definite correlation has been confirmed [15,16]. Although some studies have proposed a multiple-marker prediction model [17,18], the proposed formula was too sophisticated for clinical use, and the result was difficult to reproduce. Theoretically, an ideal malignancy prediction model should incorporate macroscopic and microscopic tumor presentations, including tumor image characteristics and tumor-cell-related markers. In this study, CTC was the only tumor presentation that was found to differ between benign and malignant lesions, whereby the CTC difference not only reached statistical significance (2.0 ± 2.83 versus 13.0 ± 15.07, *p* = 0.0314, Table 2) but also showed the largest area under the ROC curve (AUC: 0.8261, cutoff value: 6.7, *p* = 0.001), thus revealing its capacity for malignancy prediction. While CTC showed high specificity (100%), it also showed low sensitivity (60.9%). This means CTC is a highly specific marker with relatively low sensitivity, and with regard to malignancy prediction, this could mean missing cases because of relatively low sensitivity.

For a malignancy prediction model, components should have the highest sensitivity and best specificity. High sensitivity would include patients with suspicious lung lesions for further investigation. Best specificity would pick out only those patients who truly have malignant lung lesions. In our cohort, tumor size was identified as having the highest sensitivity, while CTC was found to have the best specificity. These findings were re-confirmed by analyzing sensitivity, specificity, positive predictive rate, and negative predictive rate of tumor presentations at reported cutoff values [2,7,8,9,10,14,15,25,33,34]. Based on these findings, a combination of tumor size with CTC could serve as a practical malignancy prediction model. The cutoff values for tumor size in our cohort and from the literature review were 1.3 and 0.7 cm, respectively. However, the receiver operating characteristic (ROC) curve in our cohort showed that 0.7 cm provided the best sensitivity and could include the most patients with suspicious lesions (Appendix A). In addition, the literature revealed that patients who presented with lesions greater than 0.7 cm would be confirmed with cancer within 1 year [2]. Therefore, we chose a tumor size of 0.7 cm as a cutoff value because of best sensitivity. The notable cutoff values of CTC counts in our cohort were 6.7 and 3, but the receiver operating characteristics curve for our cohort revealed that highest sensitivity and specificity were with a CTC count of 3 (Appendix A). Furthermore, the CTC cutoff value may vary depending on different purification methods [36]. Chen YY et al. utilized similar CTC purification methods to our study and identified that patients whose post-treatment CTC was greater or less than 3, but who developed elevated CTC in serial follow up, would have poor progression-free survival [33]. Therefore, we chose 3 as a cutoff value of the CTC count because of best sensitivity and specificity.

In order to prove the combined malignancy prediction model, logistical regression was performed, and Akaike information criterion (A.I.C.) of the combined malignancy prediction model was found to be 26.7361, which was lower than for tumor size or CTC alone. This result showed that the combined malignancy prediction model may have better predictive capacity (Table 4a). In the logistic regression, the combined malignancy prediction model showed marginally statistically significant (*p* = 0.0518) malignancy prediction capacity, compared to tumor size alone (*p* = 0.07). In addition, the calculated malignancy probability for patients who presented with tumor size ≥ 0.7 cm and CTC > 3 was 97.9%. Prompt tissue proving is recommended because the probability of malignancy is high. For those who presented with either tumor size ≥ 0.7 cm or CTC > 3, the calculated probabilities were 79.8 and 77.5%, respectively. In this clinical scenario, close surveillance is recommended at first for patients with sub-centimeter lesions because of a low risk of lymph node metastases even though the lesion is confirmed as malignant [36]. However, tissue proving is needed if a larger tumor size is identified. For patients with lesions greater than 1 cm, prompt tissue proving is recommended because if the lesion is confirmed as malignant, the risk of lymph node metastases is 16.3% [37]. For patients presenting with tumor size < 0.7 cm and CTC ≤ 3, the calculated probability of malignancy is 22.5%. Regular surveillance may be recommended, because the risk of malignancy is relatively low. These findings could be applied in clinical practice in order to choose the right patients and avoid unnecessary surgery (Figure 2).

There were limitations to this study. First, the small sample size prohibits further subgroup analysis and further quantification. We need to enroll more patients, and further investigation is warranted. Second, operational errors could not be completely avoided, because all processes were done by human efforts in the laboratory. In order to eliminate operational errors, we not only set up standard operating procedures for CTC purification, but also repeated the same process with two specimens to ensure the quality of laboratory work by comparing the results from these two specimens. Even though limitations remain, we have confirmed a correlation of CTC to malignancy and propose a combined malignancy prediction model with predictive capacity. Third, we realized that enrolling only four cases of benign pulmonary lesions is one of the limitations. However, we attempted to propose a better biomarker (liquid biopsy using CTCs) to compare with the conventional image criteria in regard to this clinical issue (suspicious lung lesion) and provide some supporting tools to make a surgical decision. We believe that the results can provide some exploratory and proof-of-concept evidence for future studies, and although the number of cases of benign etiology is small, this small number of cases does not seem to affect the statistical significance, which suggests the value of further investigation. Fourth, because of the specificity of Epi-CAM used in this study, we could not identify non-epithelial primary tumors through our study design. On the contrary, we could change the marker to vimentin or S-100, commonly used in sarcomas to detect circulating sarcoma cells, which poses some really interesting questions, despite our not having enrolled sarcoma patients in this study. Our method using negative depletion first can prevent most suspicious cancer cell loss in the sample preparation process, which makes changing to another surface marker to capture different types of cancer cells possible.

## Figures and Tables

**Figure 1 jpm-11-00444-f001:**
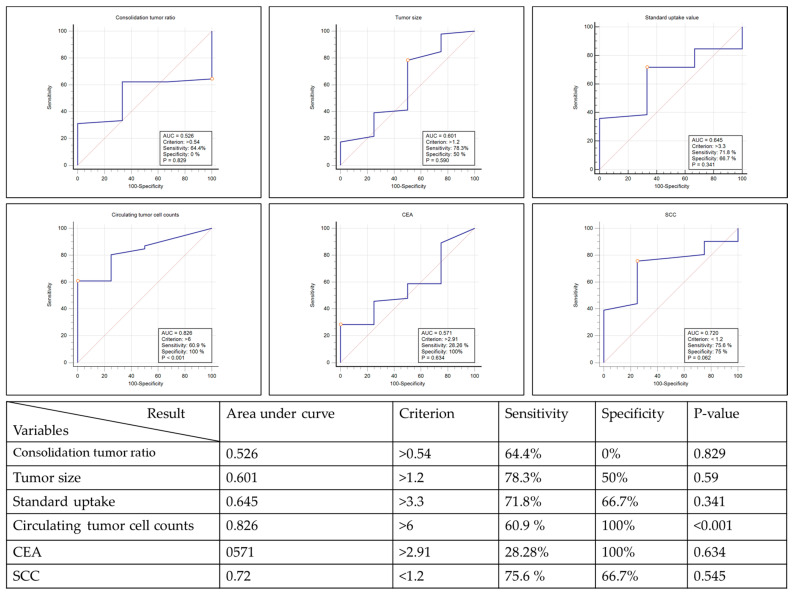
Receiver operating characteristic (ROC) curve of different tumor markers in this study cohort.

**Figure 2 jpm-11-00444-f002:**
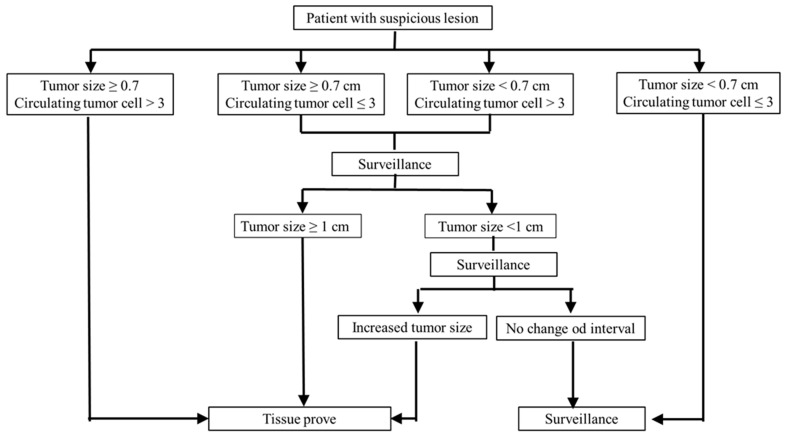
Recommended patient management algorithm based on tumor size and CTC.

**Table 1 jpm-11-00444-t001:** Characteristics of whole cohort.

Characteristics	Mean ± SD(%)	Characteristic	Mean ± SD (%)
Case number	50	PET-CT presentation	
Age (years)	64.0 ± 12.4	Dose	10.1 ± 0.6
Sex (M:F)		Blood sugar	99.2 ± 21.3
Male	27 (54%)	Tumor SUV	6.7 ± 5.3
Female	23 (46%)	Biochemical data	
ECOG scroe		Albumin	4.3 ± 0.3
0	48 (96%)	Albumin/Total protein	0.6 ± 0.1
1	2 (4%)	White blood cells	9010.0 ± 16,737.2
Smoking	16 (32%)	Seg (%)	59.9 ± 10.8
Packets per day	0.4 ± 0.8	Tumor marker (ng/mL)	
Smoking years	11.8 ± 18.5	SCC	1.1 ± 0.6
Packet years	15.9 ± 30.2	CEA	3.1 ± 3.3
PFT		CTC counts (cells/mL)	12.1 ± 14.8
FEV1	2.1 ± 0.7	Operation method	
FVC	2.6 ± 0.8	Lobectomy	29 (58%)
FEV1/FVC (%)	79.9 ± 9.1	Segmentectomy	17 (34%)
C.T. presentation		Wedge resection	4 (8%)
Tumor location		Operation times (min)	224.0 ± 63.0
Left lower lobe	5 (10%)	Blood loss (ml)	61.1 ± 83.2
Left upper lobe	11 (22%)	Pathology	
Right lower lobe	14 (28%)	Benign	4 (8%)
Right middle lobe	6 (12%)	Malignant	
Right upper lobe	14 (28%)	Lung primary	41 (82%)
Maximal tumor size	2.3 ± 1.2	Adenocarcinoma	33 (66%)
Consolidation–tumor ratio (C/T ratio)	0.7 ± 0.4	Invasive mucinous adenocarcinoma	3 (6%)
Tumor composition		Squamous cell carcinoma	3 (6%)
Pure GGO (C/T ratio:0)	9 (18%)	Other	2 (4%)
GGO predominant (CT ratio 1~50%)	3 (6%)	Metastatic	5 (10%)
Solid predominant (CT ratio: 51~99%)	26 (52%)	Tumor size (cm)	2.3 ± 1.2
Pure solid (C/Tratio:1)	10 (20%)	Hospital stay (Days)	4.2 ± 2.1
Cavitary lesion	2 (4%)		

ECOG: Estern cooperative oncology score; FEV1: Forced expiratory volume in 1 seond; FVC: Forced vital capacity.

**Table 2 jpm-11-00444-t002:** Comparative presentations of benign and malignant pulmonary lesions.

	Etiology	Benign (4)	Malignant (46)		*p*-Value *	Benign (4)	Malignant (46)	*p*-Value ^#^
Factor			Metastatic (5)	Lung Cancer (41)				
Consolidation–tumor ratio		0.7 ± 0.1 (3) ^1^	0.9 ± 0.2 (5)	0.6 ± 0.4 (40) ^3^	0.22	0.7 ± 0.2 (3) ^1^	0.6 ± 0.4 (45) ^3^	0.88
Tumor size		1.9 ± 1.1 (4)	2.8 ± 1.1 (5)	2.2 ± 1.2 (41)	0.40	1.9 ± 1.3 (4)	2.3 ± 1.2 (46)	0.51
Tumor SUV		4.2 ± 2.7 (3) ^1^	10.7 ± 5.9 (5)	6.3 ± 4.9 (35) ^4^	0.22	4.2 ± 3.3 (3) ^1^	6.9 ± 5.3 (40) ^4^	0.38
CTC		2.0 ± 2.5 (4)	17.4 ± 16.2 (5)	12.5 ± 14.7 (41)	0.09	2.0 ± 2.8 (4)	13.0 ± 15.1 (46)	0.03
CEA		1.6 ± 1.0 (4)	2.9 ± 3.2(5)	2.9 ± 3.3 (41)	0.87	1.7 ± 1.0 (4)	2.9 ± 3.3 (46)	0.64
SCC		1.4 ± 0.4 (4)	0.9 ± 0.2 (4) ^2^	1.1 ± 0.6 (37) ^5^	0.35	1.4 ± 0.4 (4)	1.0 ± 0.6 (41) ^2,5^	0.15

* Kruskal–Wallis test. ^#^ Mann–Whitney U test. ^1^ One patient, who was diagnosed with necrotizing granulomatous inflammation, presented with cavitary lesion that led to difficulty in measuring consolidation–tumor ratio and tumor standard uptake value. ^2^ One patient, who was suspected with metastatic colon adenocarcinoma, did not receive SCC check prior to tumor resection. ^3^ One patient presented with cavitary lesion that led to difficulty in measuring consolidation–tumor ratio. ^4^ Six patients received tumor scan without standard uptake value. ^5^ Four patients did not receive SCC check prior to operation.

**Table 3 jpm-11-00444-t003:** Positive predictive and negative predictive rate among different tumor markers based on cutoff values clarified in the literature.

Marker	C/T Ratio	Tumor Size	Tumor SUV	CTC	CEA	SCC
Cutoff value	>50%	≥0.7 cm	>2.5	>3 cells/mL	>3.4 ng/mL	>3.5 ng/mL
Sensitivity (95% CI)	0.73 (0.60~0.86)	0.96 (0.84~0.99)	0.78 (0.61~0.89)	0.70 (0.56~0.83)	0.78 (0.63~0.89)	1.00 (1.00~1.00)
Specificity (95% CI)	0 (0~0.69)	0.25 (0.01~0.78)	0.33 (0.02~0.87)	0.75 (0.33~1.0000)	0 (0~0.60)	0 (0~0.60)
Positive likelihood ratio (95% CI)	0.73 (0.60~0.86)	1.28 (0.72~2.25)	1.16 (0.51~2.63)	2.78 (0.50~15.36)	0.78 (0.67~0.91)	1 (1.00~1.00)
Negative likelihood ratio (95% CI)	Infinity	0.17 (0.01~2.93)	0.68 (0.10~4.43)	0.41 (0.22~0.74)	Infinity	0
Positive predictive value (95% CI)	0.92 (0.83~1.00)	0.94 (0.81~0.98)	0.94 (0.78~0.99)	0.97 (0.91~1.00)	0.90 (0.75~0.97)	0.91 (0.83~0.99)
Negative predictive value (95% CI)	0 (0~0.30)	0.33 (0.02~0.87)	0.10 (0.01~0.46)	0.18 (0.00~0.36	0 (0~0.34)	0
Accuracy (95% CI)	0.69 (0.56~0.82)	0.90 (0.82~0.98)	0.74 (0.61~0.87)	0.70 (0.57~0.83)	0.72 (0.60~0.84)	0.91 (0.83~0.99)

**Table 4 jpm-11-00444-t004:** Logistic regression for model selection.

a. Comparison of curve fitting criteria (tumor size/circulating tumor cell/combination)					
	Factors	Tumor Size ≥ 0.7 cm	Circulating Tumor Cell > 3		Combined
Curve Fitting Criteria					
Akaike information criterion (AIC)		29.21	26.74		26.73
**b. Logistic regression for model selection**					
**Model Selection**		**Odds Ratio (** **95% Confidence Interval)**		**Chi-Square (*p* Value)**	
Tumor ≥ 0.7 cm		15.00 (0.74~303.74)		0.077	
Tumor < 0.7 cm		1			
CTC > 3		12.33 (1.14~132.93)		0.038	
CTC ≤ 3		1			
Tumor ≥ 0.7 cm (controlled CTC > 3)		13.58 (0.38~484.48)		0.152	
CTC > 3 (controlled tumor size ≥ 0.7 cm)		11.85 (0.98~143.22)		0.051	

## Data Availability

Data is contained within the article and Appendix A.

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
