# Peer review of "Malignancy Prediction Capacity and Possible Prediction Model of Circulating Tumor Cells for Suspicious Pulmonary Lesions"

_jpm, 2021, doi:10.3390/jpm11060444_

Round 1

Reviewer 1 Report

Thank you for giving me the opportunity to review this manuscript.

The aim of the work is to establish a malignancy prediction model. The authors suggest that CTC might be useful for the management of lung tumor and the prediction of malignancy.

This manuscript is interesting, but the work is based on few patients, and the group of patients with benign lesions is very small (4 patients). Although the idea is interesting, I think that no definitive conclusion is brought by this work.

Minor comments:

Pathologic diagnoses should be detailed.

Major comments:

The main issue I that the authors compare 4 benign cases to 46 malignant tumors (Table 2). I am not sure that it is relevant to compare two groups having such a difference between the number of patients. Furthermore, I am not sure that it is relevant to provide a mean value on four patients.

I have the same comment for Table 3, I am not sure that calculating sensitivity and specificity values including data from a group with only 4 patients is relevant.

The proposition of an algorithm (Fig 2) is nice, but relies on data acquired on too few patients in each subgroup an is not relevant in the context.

I could not open the supplementary file .rar with the adequate software. Is the file corrupted?

Author Response

Reviewer 1

The aim of the work is to establish a malignancy prediction model. The authors suggest that CTC might be useful for the management of lung tumor and the prediction of malignancy.

This manuscript is interesting, but the work is based on few patients, and the group of patients with benign lesions is very small (4 patients). Although the idea is interesting, I think that no definitive conclusion is brought by this work.

Thanks for your comments

   Although the case numbers were limited, we utilized nonparametric statistics, including the Kruskal-Wallis test and Mann-Whitney U test to analyze the correlation between patients with malignant lesions and those with benign lesions. For people with a suspicious lung lesion requiring surgical removal, the percentage of benign pathological etiology would be relatively low in the whole study population.  Kruskal-Wallis test was utilized to analyze the correlation between primary lung malignancy, extrapulmonary malignancy with solitary metastasis and benign lesions. (Table 2)   We found that patients who presented with primary lung malignancy, extrapulmonary malignancy with solitary metastasis has higher CTC level than those with benign lesions but showed no significant statistical difference among these groups. ( p=0.09)    Therefore, we utilized the Mann-Whitney U test to analyze the difference between patients with benign and malignant lesions.  We clarify that patients with malignant lesions had higher CTC level than benign lesions with a significant statistical difference. ( p =0.03)  Other tumor macroscopic and microscopic presentations did not reveal differences among these two groups.   In addition, we further analyze the correlation between tumor presentation and tumor nature with Receiver operating characteristic (ROC) curve.  Only CTC showed a high area under the curve (AUC) and was significantly correlated with tumor nature. (p < 0.001)   According to our results, CTC showed its capability in malignancy prediction even the case numbers were limited. This study, to the best of our knowledge, is the first study to address the correlation with diagnostic images before pathology proof. If these findings can be independently validated in another prospective study they might help clinicians to decide for or against surgery

    In clinical practice, tumor size was the only criteria that guiding management for undetermined pulmonary nodules.   However, the possibility of benign etiology cannot be entirely excluded, and the risk of unnecessary operation remains.  The image size criteria have their obvious unmet needs. Many studies have attempted to find biomarkers or use a scoring system to eliminate unnecessary surgical risk. This study also calculated the positive predictive and negative predictive rates among tumor macroscopic and microscopic presentations, and found that CTC had the highest positive predictive value and specificity, compared to any other identifiable risk factors. Therefore, utilizing multiple factor analysis, we reconfirmed the malignancy prediction capacity of CTC.

Furthermore, we analyzed the efficacy of tumor size, CTC, and combined malignancy prediction model composed of tumor size and CTC by Akaike information criterion (AIC) and logistic regression. We reconfirmed the malignancy prediction capacity of CTC (p =0.04) and clarified that CTC minimizes the possibility of false-positives in the malignant prediction model based on tumor size (p-0.05). In addition, the Akaike information criterion was smaller in the combined malignancy prediction model (AIC =26.73). Based on the findings, we confirmed the malignancy prediction ability of CTC and proposed the management algorithm based on tumor size and CTC, as shown in Figure 2.

Major comments:

The main issue I that the authors compare 4 benign cases to 46 malignant tumors (Table 2). I am not sure that it is relevant to compare two groups having such a difference between the number of patients. Furthermore, I am not sure that it is relevant to provide a mean value on four patients.

I have the same comment for Table 3, I am not sure that calculating sensitivity and specificity values including data from a group with only 4 patients is relevant.

The proposition of an algorithm (Fig 2) is nice, but relies on data acquired on too few patients in each subgroup an is not relevant in the context.

I could not open the supplementary file .rar with the adequate software. Is the file corrupted?

Thanks for your comments

a.      We have addressed this issue above. We agree that having only four cases of benign pulmonary lesions is the primary limitation. However, we have attempted to propose a better biomarker (liquid biopsy using CTCs) to compare with the conventional image criteria regarding this clinical issue (suspicious lung lesion), and to provide some supporting tools for decision making. We believe that the study can provide some exploratory and proof-of-concept evidence for further studies, despite the number of cases of benign etiology being small. As we explained earlier, this number does not affect the statistical significance, and hence we calculated the mean value, sensitivity, and specificity.

b.      We have added one paragraph in lines 342-350 of page 12 to make the limitations clear to readers.

c.      We have uploaded an unzipped file for reference and confirmed that the file could open normally.

Minor comments:

Pathologic diagnoses should be detailed.

Thanks for your comments

We have provided the pathologic diagnosis in Table 1.

Reviewer 2 Report

I compliment you on your interesting study.

I have only two questions/concerns:

1) Why did you start a surveillance program for patients with proven benign resected lesions?

2) Dealing with CTC (circulating neoplastic epithelial cells) and considering metastatic lesions to the lung, I was wondering if you operated on metastases form non-epithelial primary tumours (eg: sarcomas) and, if so, which relation you may find between CTC and non epithelial primary tumour.

Author Response

Reviewer 2

I compliment you on your interesting study.

Thanks for your comments

I have only two questions/concerns:

1) Why did you start a surveillance program for patients with proven benign resected lesions?

Thanks for your comments

 We did not arrange surveillance for patients who were confirmed benign. We arranged surveillance for patients who presented with undetermined lesions with low malignancy risk, as the guidelines recommend.

2) Dealing with CTC (circulating neoplastic epithelial cells) and considering metastatic lesions to the lung, I was wondering if you operated on metastases form non-epithelial primary tumours (eg: sarcomas) and, if so, which relation you may find between CTC and non epithelial primary tumour.

Thanks for your comments.

    The reviewer has asked a very fundamental and essential question.  In our study, five patients presented with extrapulmonary malignancies, including cervical, colon, and breast cancer, with solitary pulmonary lesions and who received tumor resection. The primary sites of these patients were primary epithelial tumors. Indeed, by the design of the CTC identification of epithelial tumors as described in this study, a sarcoma would be challenging to detect.

Because of the specificity of Epi-CAM used in this study, we could not identify non-epithelial primary tumors in our study design. On the contrary, we could change the marker to vimentin or S-100, commonly used in sarcomas to detect circulating sarcoma cells, which, while we did not enroll sarcoma patients in this study, does pose some really interesting questions. Our method using negative depletion first prevented most suspicious cancer cell loss during the sample preparation process, which makes it possible to change to another surface marker in order to capture different types of cancer cells. With respect to the reviewer, we have added one paragraph in the section of the limitation in the lines of 350-357 on pages 12 and 13.

Reviewer 3 Report

Lines 113 and114 -C/T and SUV - I do not understand what these numbers are. This should be re-written for clarity.

Values in the tables can be rounded off to whole number and 1 number after the decimal without losing information and making the tables less congested. Figure 2 only needs the internal two 2 choices. I am not understanding why there is the duplication on the left and right.

In general, all the charts and figures can be less congested. The premise is correct - when CTC are detected in the peripheral blood it is indicative of metastatic disease, not just localized disease.

The concept is good; being able to have another indicator to combine with the scans of lung nodules is important to differentiate a possible malignancy from a non-malignant mass.

Overall the paper needs to be cleaned up, less congested.

Author Response

Reviewer 3

Lines 113 and114 -C/T and SUV - I do not understand what these numbers are. This should be re-written for clarity.

Thank for your comments.

  C/T ratio is the abbreviation of consolidation-tumor ratio. SUV is the abbreviation of standard uptake value. All these abbreviations are listed in the summary of abbreviations before the abstract. We have marked the changes with red color.

Values in the tables can be rounded off to whole number and 1 number after the decimal without losing information and making the tables less congested.

Thanks for your comments

We’ve modified Table 1 as per your comments. However, we only simplified Tables 2, 3 and 4 in order to preserve the information.

Figure 2 only needs the internal two 2 choices. I am not understanding why there is the duplication on the left and right.

Thanks for your comments.

These were type errors, and we have corrected following your brilliant suggestion.

In general, all the charts and figures can be less congested. The premise is correct - when CTC are detected in the peripheral blood it is indicative of metastatic disease, not just localized disease.

The concept is good; being able to have another indicator to combine with the scans of lung nodules is important to differentiate a possible malignancy from a non-malignant mass.

Thanks for your comments.

    We have simplified following your comments in order to reduce congestion. In our study, we not only identified the relationship between circulating tumor cells and malignancy but also identified its possible role in malignancy prediction in combination with tumor size.  The proposed combination malignancy model may provide more information on undetermined lung nodules and reserve tumor resection for patients who present as high risk in order to avoid unnecessary operation for benign lesions.

Overall the paper needs to be cleaned up, less congested.

Thanks for your comments

  We’ve simplified the number and table as recommended. An extensive English revision will be made again to clean up the whole text. 

Round 2

Reviewer 1 Report

I do not have any new comment.

The two other reviewers have recommended to accept this manuscript with minor revision.

I still think that the major limit of this study is the low number of patient in the control group. The authors now discuss this issue in the discussion section.

I do not have any new comment.
